# Anthropogenic and Climate Effects on a Free Dam Tropical River: Measuring the Contributions on Flow Regime

**Verônica Bernardes de Souza Léo** [1,2,*], **Hersília de Andrade e Santos** [1],
**Letícia Cristina Oliveira Pereira** [3] **and Lilia Maria de Oliveira** [4]

[1] Post-Graduation Program in Civil Engineering, Campus II, Federal Center of Tecnological Education of Minas Gerais, Minas Gerais 30510-000, Brazil; hsantos@cefetmg.br

[2] Axis of Sustainability and Environmental Issues, Campus Santa Luzia, Federal Institute of Minas Gerais, Santa Luzia 33115-390, Brazil

[3] Departament of Civil Engineering, Campus II, Federal Center of Tecnological Education of Minas Gerais, Minas Gerais 30510-000, Brazil; le.olyveirah@gmail.com

[4] Department of Environmental Science and Technology, Campus I, Federal Center of Tecnological Education of Minas Gerais, Minas Gerais 30421-169, Brazil; lilia@cefetmg.br

* Correspondence: veronica.leo@ifmg.edu.br

**Abstract:** The demand for freshwater resources and climate change pose a simultaneous threat to rivers. Those impacts are often analyzed separately, and some human impacts are widely evaluated in river dynamics—especially in downstream areas rather than the consequences of land cover changes in headwater reaches. The distinction between anthropogenic and climate on the components of the flow regime is proposed here for an upstream free dam reach whose watershed is responsible for the water supply in Rio de Janeiro. Indicators of hydrologic alteration (IHA) and the range of variability approach (RVA) combined with statistical analyses of anthropogenic and climate parameters indicated that (1) four river flow components (magnitude, frequency, duration, and rate of change) were greatly altered from the previous period (1947 to 1967) and the actual (1994 to 2014); (2) shifts in the sea surface temperature of the Atlantic correlated with flow magnitude; (3) the cattle activity effects on the flow regime of the studied area decreased 42.6% of superficial discharge; global climate change led to a 10.8% reduction in the same river component. This research indicated that climate change will impact the intensification of human actions on rivers in the southeast Brazilian headwaters.

**Keywords:** flow regime indicators; tropical headwaters; climate change; human impact

## 1. Introduction

Poor management of water resources coupled with the increased water demand for general purposes (urban supply, irrigation, animal feed, or energy production) is a human factor that directly impacts river flow [1,2]. Climate change also contributes to the significant alteration of precipitation patterns and potential evaporation [3] and consequently on river regimes [4–6]. Although human activities and climate change act collectively, their impacts on rivers dynamic are often analyzed separately [7].

Studies about the climatic and human influence on the fluvial regime of tropical rivers have recently increased. In terms of human impacts, there are analyses related to land cover changes: (a) the effects of large-scale changes in land cover on the discharge of the Tocantins River (Brazil) based on field experiments [8]; (b) a distributed hydrology soil vegetation model to analyze the effects of possible land cover change scenarios in an Atlantic Forest watershed (Brazil) [9]; (c) analyses related to dam

effects including the effect of dam cascades on the fluvial regime of the São Francisco river (Brazil) [10]. Other approaches to quantify the human effects remove climatic effects (in the São Francisco river [11] and the Mekong river [7]) or preview the effects of climate change and social-economic changes to establish the hydrological baseline for the development of an environmental flow regime (wetlands of the Eastern Zambezi [12]).

The common approaches for distinguishing the impact of climate variability and human activities on streamflow [13] used hydrological modeling [14–17], climate elasticity [18], and a hydrological sensitivity method [19]. Methods based on field data also allow the quantification of runoff alteration by different factors, and their application is the simplest. However, they demand long-term pre-impact streamflow data [20]. Changes in the natural river dynamics directly affect the biodiversity of aquatic ecosystems, and some common impact examples are the intensification of exotic species appearance and loss of local species. Social aspects include alterations of the fluvial regime that may lead to a loss of local activities by the people who depend on watercourse [21,22].

The river regime contains five main components (magnitude, frequency, duration, temporality, and rate of change) that influence the ecological integrity of the system directly or indirectly [23]. Therefore, the maintenance of the aquatic ecosystem integrity depends on the preservation of the river natural dynamics, and scientific understanding of flow-ecological relationships is crucial to guiding the setting of meaningful flow targets to restore regulated rivers or to sustaining the conservation values of rivers where new water developments are planned [24,25].

In recent decades, several approaches have proposed river dynamic evaluations [25], and the indicators of hydrologic alteration (IHA) are among the most popular metrics. These are widely used to characterize the main components of the regime. The IHA is the input for the range of variability approach (RVA), which can quantify hydrologic changes [26] and suggests that the 25th and 75th percentiles of IHA metrics should be the targets for maintaining the river dynamics similar to the natural flow regime [27].

The majority of studies for separating the impact of climate variability and human activities on streamflow evaluated these effects just under one component of flow regime: the magnitude. Thus, we aimed to evaluate the anthropogenic and climate effects on the free dam reach of a tropical river and measured their contributions to all components of the flow regime. A time-trend method was proposed based on the well-known and widely applied RVA method, and we worked to answer the following questions: (1) What are the main altered IHA metrics of the flow regime in this reach? (2) Is there a significant relationship between the main altered IHA metrics and parameters of global climate/anthropogenic activities? (3) If there is a significant relationship, what is the contribution of each parameter in the main modified components of the flow regime? A free dam river reach located in an Atlantic forest watershed was chosen to run the IHA and RVA analyses. The Preto River is a tributary of the Paraíba do Sul River, which is an essential source of water for southeast Brazil. The river regime was analyzed in two different periods: from 1947 to 1967 and from 1994 to 2014. Despite this river's importance on the water supply for states such as São Paulo and Rio de Janeiro, there are few studies about the environmental flow regime of this basin [28–31]. Only a few studies have evaluated the effects of climate change on the basins of southeast Brazil [32].

Annual data about crop areas as well as the human and cattle populations in the basin were considered as anthropogenic metrics. Data of Oceanic Niño Index (ONI) and Atlantic Multidecadal Oscillation (AMO) were applied as global climate metrics. A large-scale coupled ocean-atmosphere phenomenon in the Pacific Ocean [33] is one of the main factors that affects the interannual variability of the rainfall and temperature over much of South America [34]. Regional multidecadal climate variability is related to the AMO such as Northeastern Brazilian and African Sahel rainfall, Atlantic hurricanes, and North American and European summer climate [35].

## 2. Materials and Methods

### 2.1. Study Area

The Paraíba do Sul river basin (Figure 1a) covers areas of Minas Gerais state (36.7% of the basin area), Rio de Janeiro state (39.6% basin area), and São Paulo state (23.7% basin area). It supplies water for 90% of the inhabitants in Rio de Janeiro city and industrial regions of São Paulo [36]. The Preto River is a left tributary of Paraíba do Sul and a free dam watercourse. The studied reach (Zone 23, longitude UTM 547465 m E and latitude UTM 7530596 m S, height 976 m) is classified in the fourth-order [37] and is located in Visconde de Mauá—a district of Resende municipality (Rio de Janeiro, Brazil). The reach width is about 14 m, and the drainage area of its upstream parts is approximately 130 km$^2$. The distribution of type of lands (Figure 1b) are 67.1% of forest, 12.9 and 14.1% of dirty and clean pasture, respectively, 0.4% of exposed soil, 1.8% of urban area, 3.2% of rocky outcrop, and 0.5% of water. The local geomorphological types are mountain scarps, interplanetary depressions and Cenozoic sedimentary basin trays. Then, the region is located in dominion of gneiss-magmatritic and granulite complexes, according to the classification of geological and environmental unit. There are no aquifers under the study area [38].

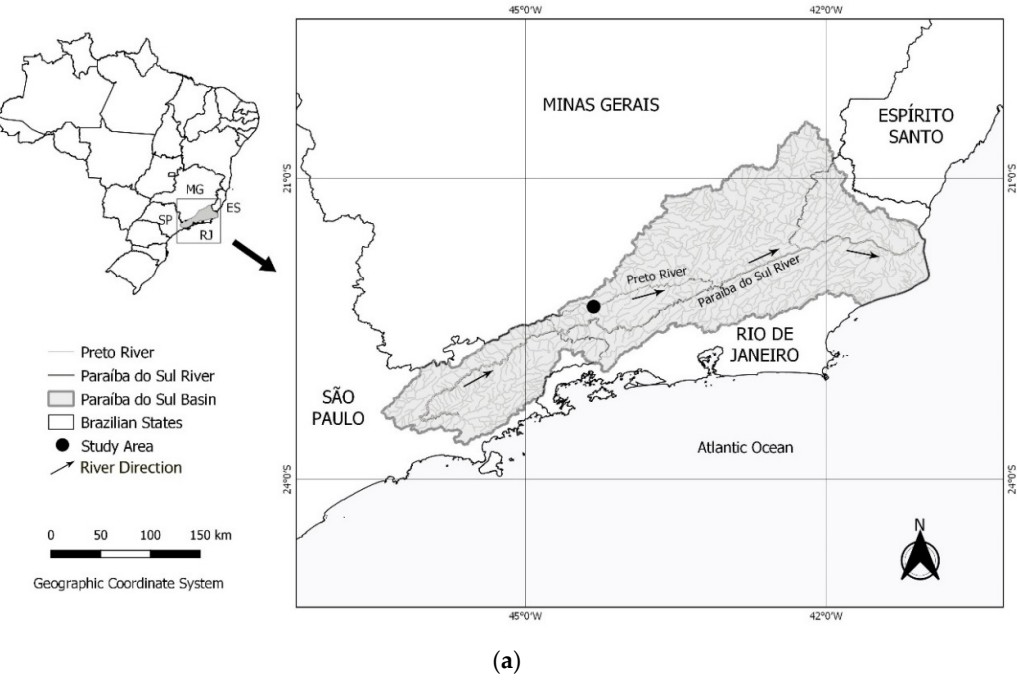

(**a**)

**Figure 1.** *Cont.*

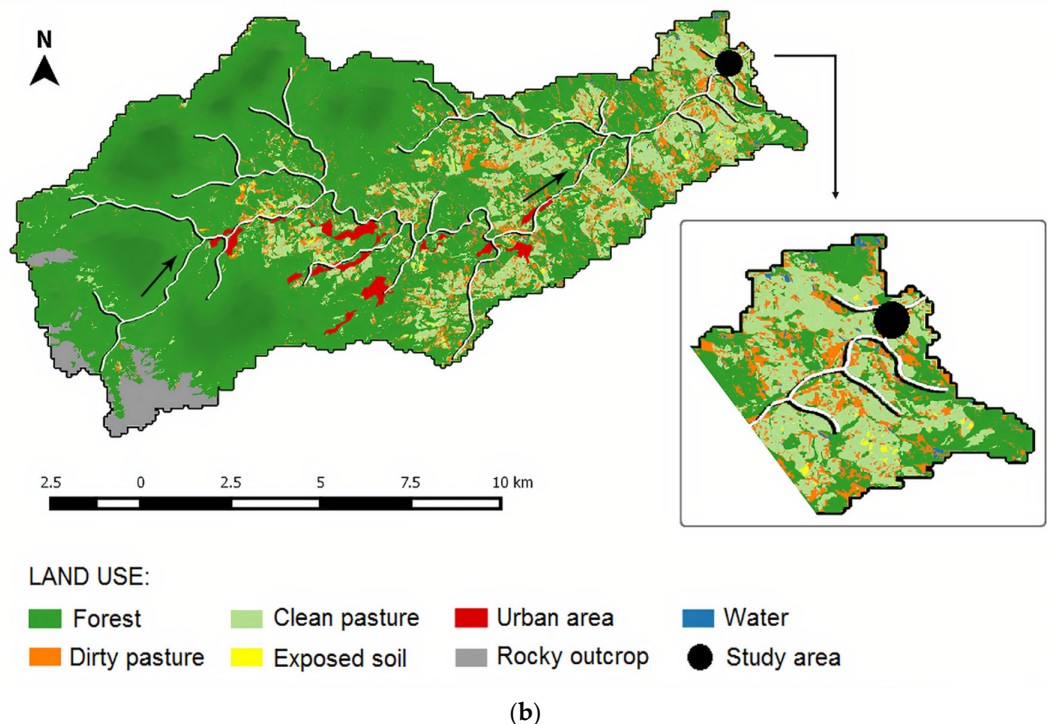

**Figure 1.** (**a**) The study area in Paraíba do Sul basin and (**b**) the land use of headwaters of studied areas. The arrows indicated the flow direction.

The study area is located in the Atlantic Forest biome, which has a high biodiversity but also many endangered species. Some parts of the Visconde de Mauá are inside the environmental protection area of Serra da Mantiqueira and the Itatiaia National Park. The local vegetation is a mix of tropical forest and mountain forests [39]. In general, the Preto river basin suffers impacts related to agricultural and livestock activities (Figure 1b) as well as urban expansion [36].

The climate is classified as subtropical highland climate [40]. From June to August, the local temperature varies from −3 °C to 6 °C. In the summer, the temperature varies between 8 °C to 27 °C. The average annual temperature is between 20 °C and 21 °C, and the average annual precipitation is around 1250 mm [41]. The rainy season is from November to April, and the dry season is from May to September.

*2.2. Indicators of Hydrologic Alteration and Range of Variability Approach*

First, 33 metrics of hydrology alteration were divided into five groups calculated using the daily discharge series obtained via a flowmeter station called Visconde de Mauá (code 5852500). The IHA calculation was run for the following times (the minimum range for RVA application is 20 years): (a) period 1: from 1947 to 1967 (the older range with 20 years); (b) period 2: from 1994 to 2014 (the most current available range with 20 years); (c) period 3: total series from 1947 to 2014 (the entire series); (d) period 4: total series from 1950–2010; (e) period 5: total series from 1974–2009 (both periods determined by analyzing of distinction of global climate and anthropogenic effects on flow magnitude—this will be explained below). The median and standard deviation of each indicator were estimated.

Second, the RVA methodology was applied [42] using the IHA from period 1 and 2: The RVA factor was calculated from the expected frequency (EF) of period 1 data and the observed frequency (OF) obtained based on period 2 data (Equation (1)). In program, the EF is calculated from the classification of values of IHA parameters found for Period 1 in three categories: (1) the lowest category that contains all values less than or equal to the 33rd percentile; (2) the middle category that contains all values

falling in the range of the 34th to 67th; (3) the highest category that contains all values greater than the 67th percentile. In this study, we used the second category. The OF is calculated in the same way, however with period 2 data.

$$RVA = \frac{(OF - EF)}{(EF * 100)}.$$ (1)

Positive RVA results indicate that the number of observations within the target range was higher in period 1 than in period 2, while negative values represent a decrease in frequency in period 2. The results are equal to zero and indicate the conservation of observations over both periods. In addition, the degree of alteration on the flow regime metric was classified in (a) low levels of alteration when RVA factor module ≤33%; (b) medium levels of alteration when RVA was between 34 and 67%; (c) high levels of alteration when RVA factor module ≥68% [42]. The software IHA available in [43] was applied for the analyses above.

### 2.3. Climate Metrics

The tendency of the RVA metrics classified at a high level of alteration were compared to the tendencies of global and local climate metrics. Global climate parameters of the Oceanic Niño Index (ONI) from 1950 to 2014 and Atlantic Multidecadal Oscillation (AMO) from 1948 to 2014 were considered for our study; both metrics were collected from the National Oceanic and Atmospheric Administration [44].

Local monthly precipitation data from 1947 to 2014 were also utilized and obtained by the rain station Visconde de Mauá (code 02244047) [45]. Daily air temperature from 1961 to 2014 was obtained from the Resende climatological station (code OMM 83738) [41].

### 2.4. Anthropogenic Metrics

Annual data about corn and bean crop area, cattle population, and human population over time were used to analyze the human influence on the river regime. These data (Table 1) were obtained from the National Institute of Geography and Statistics (IBGE) and the Institute of Applied Economic Research (IPEA) for Resende municipality. According to the frequency of the studies carried out by these Institutes, there are only data available every 5 years for the variables bean cultivation area and cattle population and around 10 years for inhabitants.

**Table 1.** Anthropogenic metrics existing to compare with discharge data of similar years: basin inhabitants, corn/bean crop areas (hectare), and cattle population for Resende city [46].

| Metric/Year | 1940 | 1950 | 1960 | 1970 | 1980 | 1991 | 1996 | 2000 | 2007 | 2010 |
|---|---|---|---|---|---|---|---|---|---|---|
| Inhabitants | 27,422 | 34,752 | 48,165 | 66,907 | 87,338 | 91,757 | 102,625 | 104,549 | 118,547 | 119,770 |

| Metric/Year | 1974 | 1979 | 1984 | 1989 | 1994 | 1999 | 2004 | 2009 | | |
|---|---|---|---|---|---|---|---|---|---|---|
| Corn/bean crop areas (hectare) | 480 | 172 | 1938 | 1580 | 1190 | 610 | 250 | 213 | | |
| Cattle population | 41,963 | 42,630 | 50,900 | 51,400 | 50,716 | 28,200 | 30,992 | 29,600 | | |

### 2.5. Statistical Analysis

The Wilcoxon signed-ranks test was applied to verify the differences in climate metrics (AMO, ONI, local temperature, monthly precipitation, annual precipitation) for period 1 and 2. A Pearson's correlation was run for a wide range of combinations to verify links between:

1. The local and global climate metrics: (a) monthly accumulated precipitation and monthly mean AMO; (b) monthly accumulated precipitation and monthly mean ONI; (c) monthly accumulated precipitation and monthly mean temperature.
2. The flow magnitude and the climate metrics: (a) monthly mean flow magnitude and monthly accumulated precipitation and (b) monthly mean flow magnitude and monthly mean AMO.

3.　The flow regime parameters with a high level of alteration in RVA (from groups 2 to 5) and the annual climate metrics: metrics of group 1 were not considered, because they are related to the monthly analyses.

4.　The flow magnitude and the anthropogenic metrics: (a) annual mean flow magnitude and annual mean inhabitants; (b) annual mean flow magnitude and annual mean crop area; (c) annual mean flow magnitude and annual mean cattle population.

For this study, Pearson's correlation, in module, between 0.00 and 0.20 indicated weak correlation, between 0.20 and 0.40 indicated low correlation, between 0.40 and 0.60 a moderate correlation, 0.60 and 0.80 a high correlation, 0.80 and 0.90 a very high correlation, and values between 0.90 and 1.00 implied almost full correlation [47]. The adopted level of significance was 10% (*p*-value equal 0.10), and all the analyses were run in R software version 4.0.2 (2020) [48].

### 2.6. Distinction of Global Climate and Anthropogenic Effects on Flow Magnitude

The annual flow magnitude and precipitation events were first classified based on their variation. According to [11], the mean annual parameter (Qm or Pm) should be set as the reference value for the annual hydrological condition, and the variations were evaluated by the standard deviation ($\sigma$) of the annual average discharge values. The limits of the different classes of hydrological events (Table 2) is determined by the value of one standard deviation. Thus, the anomaly is the variability of annual parameters around the mean by normalizing the series (Equations (2) and (3)):

$$\text{AnomalyP} = \frac{(\text{Pi} - \text{Pm})}{\sigma_p}, \tag{2}$$

$$\text{AnomalyQ} = \frac{(\text{Qi} - \text{Qm})}{\sigma_q}. \tag{3}$$

**Table 2.** Classification of the annual discharge and precipitation events based on the anomaly of annual average streamflow and anomaly of annual total precipitation.

| Limits | Classification |
| --- | --- |
| Anomaly < −1.5 | Very Dry |
| −1.5 < Anomaly < −0.5 | Dry |
| −0.5 < Anomaly < 0.5 | Average |
| 0.5 < Anomaly < 1.5 | Wet |
| Anomaly > 1.5 | Very Wet |

Here, Pi is the annual total precipitation (mm) in the year i, Pm is the mean annual total precipitation (mm) of all data series, $\sigma_p$ is the standard deviation (mm), Qi is the annual average streamflow ($m^3$/s) in the year i, Qm is the mean annual streamflow ($m^3$/s) of all data series, and $\sigma_q$ is the standard deviation ($m^3$/s). The data series corresponded to period 3 (from 1947 to 2014). For anomaly analyses, series should be without missing data for the entire reference year. That is why the calculation was performed for series of 1946–1951; 1957–1958; 1965; 1970; 1972; 1974; 1984; 1986; 1989–1991; 1994; 2004; 2007; 2012–2014.

Second, the years' anomalies in the same class for precipitation and discharge were selected to create two flow magnitude duration curves: period 1 and period 2. These curves were based on daily discharge and represent the variability and exceedance probability of flow over the available (or selected) period [49]. Both curves were compared, and the difference represents the quantification of global climate and anthropogenic effects on discharge.

Finally, the difference between the flow magnitude duration curves indicates the flow magnitude generated by no local climate effects such as global climate and anthropogenic effects.

Distinguishing each effect on flow magnitude was estimated by the determination coefficient from linear regression between the flow magnitude and the corresponding explicative variable.

## 3. Results

### 3.1. Indicators of Hydrologic Alteration and Range of Variability Approach

The IHA indicates a decrease in monthly flow magnitude in the rainy season (median deviations from −18.18 to −7.69%) and in July (−33.33%) (Table 3). The minimum and maximum annual flows of 30 days also led to a reduction (median deviations 31.67 and 24.95%, respectively) (Figure 2a,b). The frequency indicator for low pulses per year (Figure 2c) showed a high deviation from the median (400%). The number of reversals (Figure 2d) increased for period 2 (26.23%).

**Table 3.** Indicators of hydrologic alteration (IHA) for period 1 (1947–1967) and period 2 (1994–2014) as well as the range of variability approach (RVA) for a studied area. The grey cells indicate the high level of alteration via RVA analyses.

| IHA Group | IHA—Means | | | RVA | |
| | Streamflow (m³/s) | | | | |
| | **Period 1** | **Period 2** | **Deviation Magnitude (%)** | **Deviation from Target Range (%)** | **Classification** |
|---|---|---|---|---|---|
| **Group 1: Monthly magnitude** | | | | | |
| October | 2 | 2 | 0 | 9.091 | Low |
| November | 3.5 | 3 | −14.29 | 33.33 | Low |
| December | 6 | 5 | −16.67 | 112.5 | High |
| January | 8 | 7 | −12.5 | −25 | Low |
| February | 11 | 9 | −18.18 | −30 | Low |
| March | 10 | 9 | −10 | 37.5 | Average |
| April | 6.5 | 6 | −7.692 | 50 | Average |
| May | 4 | 4 | 0 | −6.667 | Low |
| June | 3 | 3 | 0 | 9.091 | Low |
| July | 3 | 2 | −33.33 | 0 | None |
| August | 2 | 2 | 0 | −6.25 | Low |
| September | 2 | 2 | 0 | −41.18 | Average |
| **Group 2: Magnitude and duration of annual extremes** | | | | | |
| 1 day minimum | 1 | 1 | 0 | 40 | Average |
| 3 day minimum | 1 | 1 | 0 | 42.86 | Average |
| 7 day minimum | 1.14 | 1 | −12.5 | 42.86 | Average |
| 30 day minimum | 2 | 1.37 | −31.67 | −76.92 | High |
| 90 day minimum | 2.23 | 1.89 | −15.42 | −85.71 | High |
| 1 day maximum | 32 | 27 | −15.63 | −87.5 | High |
| 3 day maximum | 24 | 21.33 | −11.11 | −14.29 | Low |
| 7 day maximum | 21.57 | 17.29 | −19.87 | −42.86 | Average |
| 30 day maximum | 15.37 | 11.53 | −24.95 | −85.71 | High |
| 90 day maximum | 11.11 | 9.544 | −14.1 | −71.43 | High |
| Number of zero-flow days | 0 | 0 | 0 | 0 | Average |
| Base flow (%) | 0.24 | 0.22 | −8.51 | 85.71 | High |
| **Group 3: Timing of annual extremes** | | | | | |
| Julian date of annual minimum | 275 | 275 | 0 | −18.18 | Low |
| Julian date of annual maximum | 23 | 11 | −6.56 | 14.29 | Low |

**Table 3.** *Cont.*

| IHA Group | IHA—Means | | | RVA | |
| --- | --- | --- | --- | --- | --- |
| | Streamflow (m³/s) | | | | |
| | Period 1 | Period 2 | Deviation Magnitude (%) | Deviation from Target Range (%) | Classification |
| Group 4: Frequency and duration of high and low pulses | | | | | |
| Low pulse count | 1 | 5 | 400 | −70 | High |
| Low pulse duration | 5.75 | 5 | −13.04 | 33.33 | Low |
| High pulse count | 7 | 10 | 42.86 | −22.22 | Low |
| High pulse duration | 3 | 2 | −33.33 | −25 | Low |
| Group 5: Rate and frequency of change flow | | | | | |
| Rise rate | 1.5 | 1 | −33.33 | 10.53 | Low |
| Fall rate | −1 | −1 | 0 | 5 | Low |
| No. of flow reversals | 61 | 77 | 26.23 | −100 | High |

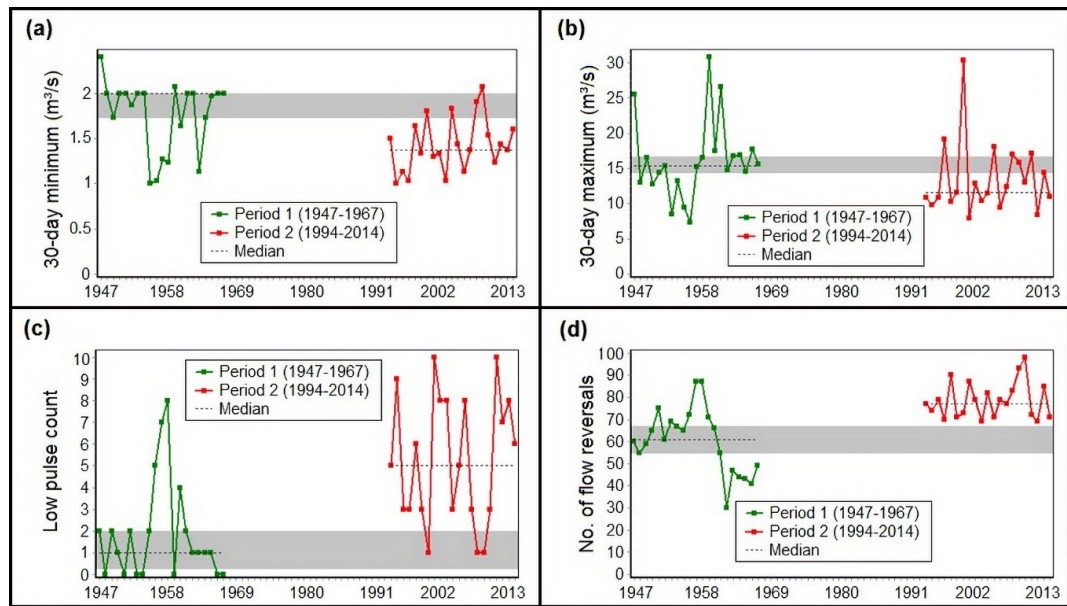

**Figure 2.** Some indicators of hydrologic alteration (IHA) along with period 1 (1947–1967) and period 2 (1994–2014), which show high levels of alteration via RVA analyses: (**a**) the minimum annual flow of 30; (**b**) maximum annual flows of 30 days; (**c**) low pulse count; (**d**) the number of flow reversals. The area in dark grey indicates the target range of the RVA method.

The degree of alteration on flow regime was classified by RVA analysis as a high level over nine metrics (Table 3): December magnitude flow (112.5% deviation from the target range); minimum annual flow of 30 and 90 days (−76.92 and −85.71%); maximum annual flows of 1, 30, and 90 days (−87.5, −85.71, and −71.43%); base flow (−85.71%); low pulses per year (−70%); and the number of reversals (−100%).

*3.2. Local and Global Climate Metrics Differences between Periods 1 and 2*

Significant changes between periods 1 and 2 occurred in the air temperature of December and AMO of June and October as well as the period median (Table 4). No significant changes were detected for local precipitation and ONI.

**Table 4.** *p*-values of Wilcoxon test for Atlantic Multidecadal Oscillation (AMO), Oceanic Niño Index (ONI), monthly mean temperature, and monthly precipitation accumulation in period 1 and period 2. The dark grey cells indicate a *p*-value lower than 10%.

| Period | Monthly Precipitation Accumulations | Monthly Mean Temperatures | Monthly Mean AMO | Monthly Mean ONI |
|---|---|---|---|---|
| January | 0.768 | 0.068 | 0.135 | 0.556 |
| February | 0.122 | 0.225 | 0.526 | 0.355 |
| March | 0.876 | 0.612 | 0.627 | 0.257 |
| April | 0.737 | 0.866 | 0.794 | 0.117 |
| May | 0.852 | 0.310 | 0.126 | 0.123 |
| June | 0.170 | 0.398 | 0.030 | 0.256 |
| July | 0.476 | 0.345 | 0.023 | 0.217 |
| August | 0.289 | 0.310 | 0.006 | 0.420 |
| September | 0.664 | 0.091 | 0.004 | 0.355 |
| October | 0.715 | 0.866 | 0.025 | 0.368 |
| November | 0.575 | 0.735 | 0.391 | 0.395 |
| December | 0.931 | 0.043 | 0.601 | 0.337 |
| Annual mean | - | 0.753 | 0.100 | 0.145 |
| Annual median | - | 0.612 | 0.057 | 0.236 |
| Annual accumulation | 0.330 | - | - | - |

### 3.3. Links between Local and Global Climate Metrics

Linear relations between the monthly accumulated precipitation (local climate metric) and monthly mean AMO (global climate metric) in periods 2 and 3 presented a *p*-value lower 0.05 but low and weak negative correlation by Pearson's analyses (Pearson's −0.276 and −0.070). A high positive correlation between local climate metrics such as monthly precipitation accumulations and monthly mean temperature parameters were obtained for periods 1, 2, and 3 (Table 5).

**Table 5.** Pearson's coefficient and *p*-values of linear regression for global and local climate metrics. The dark grey cells indicate a *p*-value lower than 5%.

| Relations | Factors | Period 1 | Period 2 | Period 3 |
|---|---|---|---|---|
| Monthly precipitation accumulation versus Monthly mean AMO | Pearson's r | 0.022 | −0.276 | −0.070 |
| | *p*-value | 0.739 | <0.05 | <0.05 |
| | $R^2$ | - | 0.076 | 0.005 |
| Monthly precipitation accumulation versus Monthly mean ONI | Pearson's r | −0.015 | −0.083 | −0.026 |
| | *p*-value | 0.825 | 0.194 | 0.469 |
| | $R^2$ | - | 0.007 | - |
| Monthly precipitation accumulation versus Monthly mean temperature | Pearson's r | 0.624 | 0.698 | 0.683 |
| | *p*-value | <0.05 | <0.05 | <0.05 |
| | $R^2$ | 0.389 | 0.487 | 0.466 |

### 3.4. Links between the Flow Magnitude and the Climate Metrics

The mean monthly flow magnitude in periods 1, 2, and 3 presented high positive correlation to monthly precipitation accumulation (Pearson's r > 0.5). A low negative correlation was seen for the mean monthly flow magnitude and the global climate metric monthly AMO (Table 6).

**Table 6.** Pearson's correlation and *p*-values of linear regression for climate metrics and the flow regime. The dark grey cells indicate a *p*-value lower than 5%.

| Relations | Factors | Period 1 | Period 2 | Period 3 |
|---|---|---|---|---|
| Monthly mean flow magnitude versus Monthly precipitation accumulation | Pearson's r | 0.740 | 0.787 | 0.763 |
| | *p*-value | <0.05 | <0.05 | <0.05 |
| | $R^2$ | 0.548 | 0.619 | 0.582 |
| Monthly mean flow magnitude versus Monthly mean AMO | Pearson's r | 0.009 | −0.328 | −0.040 |
| | *p*-value | 0.892 | <0.05 | 0.257 |
| | $R^2$ | - | 0.108 | 0.001 |

### 3.5. Links between the Flow Regime Parameters with a High Level of Alteration in RVA and the Climate Metrics

Among the six parameters with a high level of alteration in RVA and with a *p*-value lower than 5%, three showed a low correlation with the annual monthly precipitation (the minimum annual flows

of 30 days, the maximum annual flow of 1 day, and the annual base flow) and three showed moderate correlation (the maximum annual flows of 30 days and the minimum and maximum annual flows of 90 days). The linear relationship between the number of reversals and the annual median AMO presented a *p*-value lower than 10%. The number of low pulses, however, did not show a significant linear relationship with both climate metrics (Table 7).

### 3.6. Links between the Flow Regime and the Anthropogenic Metrics

A high negative correlation (Pearson's r > 0.5) between the annual flow magnitude and annual cattle population was obtained for the period from 1974 to 2009 (Table 8).

### 3.7. Links between the Flow Regime Parameters with a High Level of Alteration in RVA and the Anthropogenic Metrics

Three flow magnitude parameters with a high level of alteration in RVA had a high correlation to annual corn/bean crop areas (Pearson's r > 0.5): the maximum annual flow of 30 and 90 days and the annual base flow. Their linear relation presented a *p*-value less than 10% (Table 9). The annual cattle population can also influence the maximum annual flow of 90 days, and the number of reversals in the flow regime was strongly correlated to the annual population.

### 3.8. Distinction of Global Climate and Anthropogenic Effects on Flow Magnitude

From 1945 to 2015, the annual discharge and precipitation anomalies changed from the wet category to an average category and presented a similar linear tendency (Figure 3). However, the tendency lines suggest a reduction in the anomaly's values faster for flow magnitude than precipitation. The flow magnitude anomaly behavior does not follow the precipitation anomaly in 1951, 1957, 1994, 2004, and 2012.

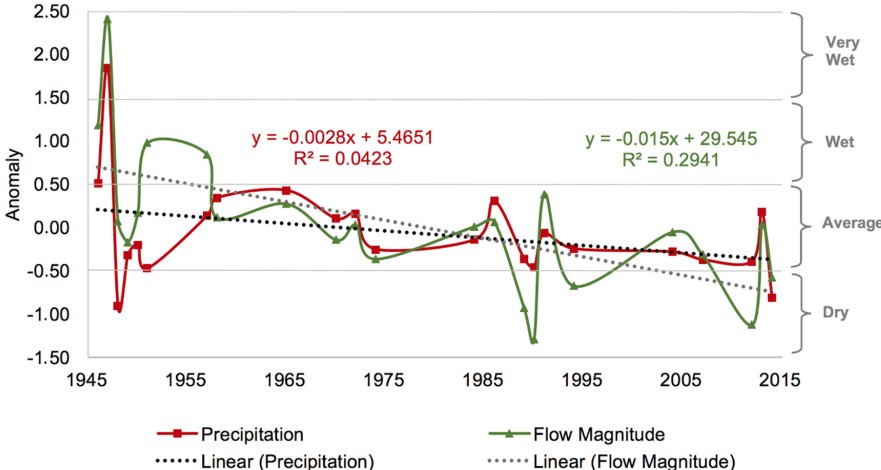

**Figure 3.** Annual discharge and precipitation anomalies from 1945 to 2015 and their classification for data from flowmeter station Visconde de Mauá (code 5852500) and pluviometric station (code 02244047).

**Table 7.** Pearson's correlation and *p*-values of linear regression for climate metrics and metrics and the flow regime parameters with high level of alteration in RVA. The dark grey cells indicate a *p*-value lower than 5%. The light grey cells indicate a *p*-value lower than 10%. The symbol # represents the word "number".

| Parameter | Factors | 30 Day min | 90 Day min | 1 Day max | 30 Day max | 90 Day max | Base Flow | Low Pulse [#] | Reversals |
|---|---|---|---|---|---|---|---|---|---|
| Annual precipitation accumulation | Pearson's r | 0.341 | 0.456 | 0.271 | 0.410 | 0.459 | −0.299 | −0.196 | 0.058 |
| | *p*-value | <0.05 | <0.05 | <0.05 | <0.05 | <0.05 | <0.05 | 0.112 | 0.644 |
| | $R^2$ | 0.116 | 0.208 | 0.073 | 0.168 | 0.211 | 0.089 | 0.038 | 0.003 |
| Annual mean AMO | Pearson's r | 0.059 | −0.018 | 0.064 | 0.119 | 0.109 | −0.049 | 0.077 | 0.220 |
| | *p*-value | 0.634 | 0.882 | 0.608 | 0.336 | 0.378 | 0.694 | 0.534 | <0.10 |
| | $R^2$ | 0.003 | - | 0.004 | 0.014 | 0.012 | 0.003 | 0.006 | 0.048 |

**Table 8.** Pearson's correlation and *p*-values of linear regression for flow magnitude and anthropogenic metrics. The light grey cells indicate a *p*-value lower than 10%.

| Relations | Analyses Period | Pearson's Correlation | *p*-Value | $R^2$ |
|---|---|---|---|---|
| Annual mean flow magnitude versus Annual population | 1940–2010 (period 4) | −0.054 | 0.890 | 0.003 |
| Annual mean flow magnitude versus Annual corn/bean crop areas | 1974–2009 (period 5) | −0.457 | 0.209 | 0.209 |
| Annual mean flow magnitude versus Annual cattle population | 1974–2009 (period 5) | −0.653 | <0.10 | 0.426 |

**Table 9.** Pearson's correlation and *p*-values of linear regression for anthropogenic metrics and the flow regime parameters with a high level of alteration in RVA. The dark grey cells indicate a *p*-value lower than 5%. The light grey cells have a *p*-value lower than 10%. The symbol # represents the word "number".

| Parameter | Factors | 30 Day min | 90 Day min | 1 Day max | 30 Day max | 90 Day max | Base Flow | Low Pulse [#] | Reversals |
|---|---|---|---|---|---|---|---|---|---|
| Annual population | Pearson's correlation | −0.486 | −0.542 | 0.070 | 0.120 | −0.113 | 0.087 | 0.177 | 0.611 |
| | *p*-value | 0.185 | 0.132 | 0.859 | 0.758 | 0.771 | 0.824 | 0.649 | <0.10 |
| | $R^2$ | 0.236 | 0.294 | 0.005 | 0.014 | 0.013 | 0.008 | 0.031 | 0.373 |
| Annual corn/bean crop areas | Pearson's correlation | −0.092 | −0.318 | −0.382 | −0.728 | −0.692 | 0.697 | −0.357 | −0.241 |
| | *p*-value | 0.829 | 0.443 | 0.351 | <0.05 | <0.10 | <0.10 | 0.386 | 0.566 |
| | $R^2$ | 0.008 | 0.101 | 0.146 | 0.530 | 0.479 | 0.486 | 0.127 | 0,058 |
| Annual cattle population | Pearson's correlation | −0.281 | −0.474 | 0.113 | −0.431 | −0.679 | 0.451 | 0.192 | −0.310 |
| | *p*-value | 0.499 | 0.236 | 0.791 | 0.286 | <0.10 | 0.262 | 0.649 | 0.455 |
| | $R^2$ | 0.079 | 0.225 | 0.013 | 0.186 | 0.461 | 0.203 | 0.037 | 0.096 |

The years from 1949 to 1965 in period 1 and the years of 1990 to 2013 in period 2 were selected to create two flow magnitude duration curves, because they are the largest data series inserted in the same anomaly category, i.e., average (Figure 4).

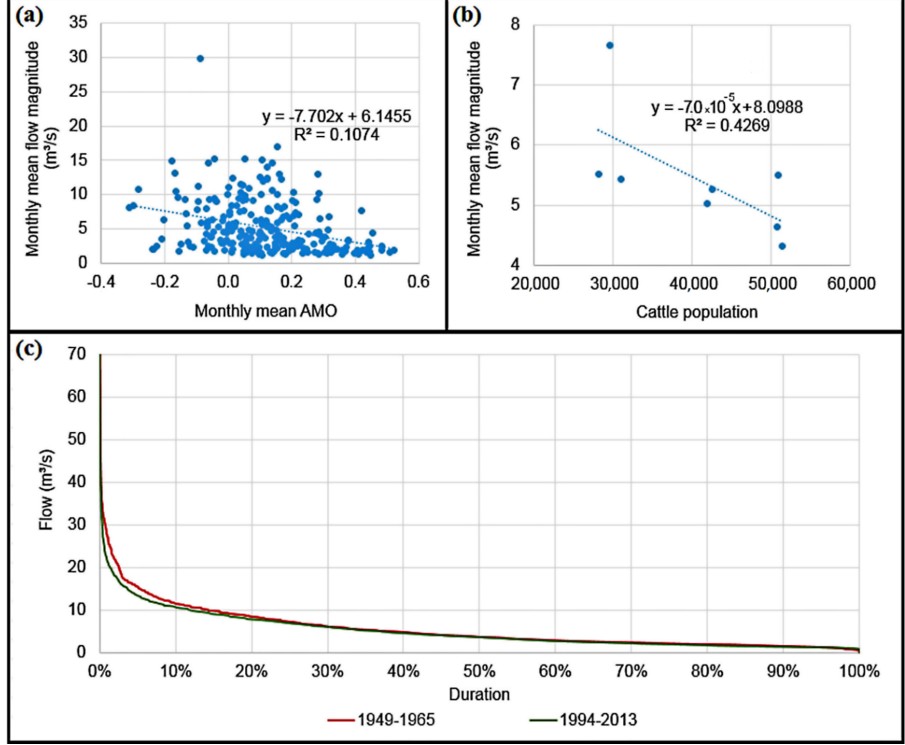

**Figure 4.** (**a**) Linear regression between the monthly mean AMO and monthly mean flow magnitude, (**b**) linear regression between the cattle population and monthly mean flow magnitude, and (**c**) flow magnitude duration curves for the average category for 1949 to 1965 (period 1) and 1994 to 2013 (period 2).

The total difference between the flow magnitude duration curves (Table 10) was $1.35 \times 10^7$ m$^3$/year (0.43 m$^3$/s). This loss of flow magnitude in period 2 was attributed partially to global climate and anthropogenic effects. Concerning the power explanation of linear regression between the metrics, 10.8% ($14.58 \times 10^5$ m$^3$/year) of this flow magnitude was related to global climate changes (indicated by AMO), and 42.6% ($57.51 \times 10^5$ m$^3$/year) was related to livestock activities

**Table 10.** The difference of flow magnitude duration curves for period 1 and period 2.

| Duration Curve | 10% | 20% | 30% | 40% | 50% | 60% | 70% | 80% | 90% | 100% |
|---|---|---|---|---|---|---|---|---|---|---|
| Flow magnitude—Period 1 (m$^3$/s) | 11.598 | 8.584 | 6.226 | 4.955 | 3.810 | 2.978 | 2.493 | 2.089 | 1.648 | 0.600 |
| Flow magnitude—Period 2 (m$^3$/s) | 10.795 | 7.943 | 6.208 | 4.690 | 3.761 | 2.843 | 2.317 | 1.838 | 1.494 | 0.800 |
| % Flow alteration | −6.9% | −7.5% | −0.3% | −5.4% | −1.3% | −4.5% | −7.0% | −12.0% | −9.3% | 33.3% |

## 4. Discussion

The effects of human activities on streamflow regimes have been quantified by several methods, and RVA was widely applied to determine the impacts of dam construction in rivers around the world [50–53]. However, the evolution of other anthropogenic effects such as the influence of land cover and water withdraws on river runoff by RVA is rare especially in headwater reaches due to the

lack of long hydrologic records in those areas. This is necessary to reduce the interference of climate variability on the RVA method. Hydrological models were also used in approaches to estimate the contribution of climate change and human activities (especially the reservoir operations) on flow changes [14–17]. Few studies have correlated the measured components of flow regime to climate and local land cover indicators.

The flow regime of the studied headwater and free dam reach of the Preto River presented high altered IHA in four river components (magnitude, frequency, duration, and rate of change) and some flow regime metrics: the minimum annual flow was 30 and 90 days; maximum annual flows of 1, 30, and 90 days; base flow; the number of reversals). The metrics were correlated with some anthropogenic or global climate metric or even to both metrics. Shifts in the Atlantic Ocean behavior were significantly different between the analyzed time-series and the most altered metrics of flow regime that correlated strongly with AMO in the most current period. Our results implied the cattle activity effects on flow regime during this period decreased 42.6% of superficial discharge, and global climate changes lead to a 10.8% reduction in the same river component.

The climate patterns of the Pacific Ocean did not alter the variability of the rainfall and temperature in this reach of the Preto River, although the El Nino effects are widely described in other parts of the Brazilian Southeast [54,55] and even for downstream areas of the Paraíba do Sul basin [56]. The effects of temperatures variation in the North Atlantic sea surface, which is described in the literature as an influence factor for rainfall in the North Eastern Brazil [35], altered the accumulated monthly precipitation in the most current studied period and became an important driver to the changes in magnitude or numbers of reversals in the flow regime of this free dam reach. The strong correlations of AMO with discharge were observed at headwater catchments in the Northeastern United States [57], and further comparisons with the circulation patterns of the South Atlantic Ocean may indicate the strongest relation between the sea temperature in this ocean and the headwaters of Brazilian Southeast.

More severe changes in the Atlantic temperatures can lead to losses of 10% of the available flow in this section. Recent studies on the scale of the large basins projected the effects of climate change on the discharge of South America rivers, and decreases in the annual mean discharge are expected in the larges watershed of Brazilian Southeast (São Francisco, Paraná, and Doce) despite the lack of agreement of global climate models over most of this region [58].

Although the AMO explained rainfall variation for 1994 to 2014, no significant differences in monthly total precipitation between the older and current periods were observed. However, the precipitation anomalies declined along with the entire studied series (from 1947 to 2014), and the discharge anomalies followed this tendency. Larges rivers in the Brazilian Southeast (Paraguaçu, Paraíba do Sul, de Contas, and Doce) presented decreased discharges from 1985 at the mouth of each respective watershed [56].

Human activities such as land cover alterations and water withdrawal are also responsible for changes in hydrological components and discharge alterations [23]. In other studies, the percentages in the change in runoff caused by human activity were higher than 47% [17]. The cattle population of the studied area correlated strongly with the mean flow magnitude and the growth of cattle inhabitants in the area produced in recent years may decline water availability by $57.51 \times 10^5$ m$^3$/year. Water demands by livestock during its life cycle may disturb superficial water quantities in an area and may create damage to the environment [59]. Research into commodity production suggests that water importation and exportation in products from crops and the meat products contributed in 90 km$^3$year$^{-1}$ for global virtual water exports in 2000, which came mainly from streamflow in case of Brazilian meat production [60].

In South America, livestock is the second biggest consumer of water [61] and, together with other land-use practices, such as agriculture and urbanization, is likely the primary cause of altered flow regimes in headwaters rather than dams. However, climate change may worsen those impacts on river dynamics, which have an ecological function [62] and therefore on water availability for human consumption or ecosystem processes. Similar studies in China have also shown the impact of

intensification of human activities on river runoff by climate change [15,17]. Changes in the number of flow reversal metrics obtained here also indicated the influence of livestock activities and climate change on the other two flow regime components: frequency and rate of change.

Finally, other anthropogenic activities may contribute to the impact that the flow regime has on the headwaters of the Brazilian Southeast such as urbanization and tourism; those influences may increase in the coming years due to the hydrologic stress caused by water pollution and water scarcity in populated downstream areas (in the case of Rio Preto, the downstream area is part of Rio de Janeiro metropole). Further studies should consider the combined impact of these anthropogenic activities on river dynamics as well as their future intensification from climate change. Our proposed time-trend method for separating the impact of climate variability and human activities on flow regime could be applied in these other analyses and enlargement of input data may improve the method results.

## 5. Conclusions

Indicators of hydrologic alteration (IHA) and the range of variability approach (RVA) were combined for statistical analyses of anthropogenic and climate parameters and applied for a free-dam upstream reach in the Brazilian Southeast. These metrics are useful to distinguish anthropogenic from climate contributions on the components of the flow regime.

The flow regime of tropical headwaters is highly altered in four river components (magnitude, frequency, duration, and rate of change) by human activity. These impacts are intensified by recent climate changes especially in the flow discharge (10.8% reduction in this flow component). The livestock may play an important role in the alterations of the flow magnitude (42.6% reduction) and the rate of change in tropical rivers' sources, especially in countries whose economy is based on commodity exportation. Thus, this study could deepen the understanding of interactive effects of human activity and climate change on tropical headwaters.

**Author Contributions:** Conceptualization, V.B.d.S.L. and H.d.A.e.S.; methodology, V.B.d.S.L. and H.d.A.e.S.; software, V.B.d.S.L., H.d.A.e.S., L.C.O.P. and L.M.d.O.; validation, V.B.d.S.L. and H.d.A.e.S.; formal analysis, V.B.d.S.L. and H.d.A.e.S.; investigation, V.B.d.S.L. and H.d.A.e.S.; resources, V.B.d.S.L. and H.d.A.e.S.; data curation, V.B.d.S.L., H.d.A.eS., L.C.O.P., and L.M.d.O.; writing—original draft preparation, V.B.d.S.L. and H.d.A.e.S.; writing—review and editing, V.B.d.S.L. and H.d.A.e.S.; visualization, V.B.d.S.L., H.d.A.e.S., L.C.O.P. and L.M.d.O.; supervision, H.d.A.e.S.; project administration, H.d.A.e.S.; funding acquisition, H.d.A.e.S. All authors have read and agreed to the published version of the manuscript.

**Funding:** This research was funded by Centro Federal de Educação Tecnológica de Minas Gerais—CEFET-MG and Fundação de Amparo à Pesquisa do Estado de Minas Gerais—FAPEMIG (process number APQ-01145-15).

**Acknowledgments:** The authors are grateful to the Centro Federal de Educação Tecnológica de Minas Gerais (CEFET-MG) for supporting the student scholarships of V.B.d.S.L. and Instituto Federal de Minas Gerais campus Santa Luzia for supporting the study leave of V.B.d.S.L. The authors also thank Fundação de Amparo à Pesquisa do Estado de Minas Gerais (FAPEMIG) for its financial support through the project "Vazão ecológica para trecho da bacia do Paraíba do Sul: uma abordagem com multivariáveis biológicas" (process number APQ-01145-15). This study was also partly funded by Coordenação de Aperfeiçoamento de Pessoal de Nível Superior, Brazil (CAPES), Finance Code 001. The authors are grateful to Itamar Gonçalves, Ivan Batista, and Gilbete Santos for support in the field research and in the computational laboratory, and the staff of Parque Estadual da Pedra Selada for providing lodgings during the field samples.

**Conflicts of Interest:** The authors declare no conflict of interest.

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
