# Peer review of "Anthropogenic and Climate Effects on a Free Dam Tropical River: Measuring the Contributions on Flow Regime"

_sustainability, doi:10.3390/su122310030_

Round 1
Reviewer 1 Report
Minor observations
Line 105: Announce Figure 1b before the figure.
Line 181: Check “1σ”.
Line 122: Announce Table 4 before the table.
Line 231: In “p-value” use “P”.
Line 267, Figure 3: Draw the graphical representation as in Figure 2 and not choosing “smoothed”.
Line 280: Use “.” at the end of the phrase.
General remarks
Excellent paper! Congratulations!
Reviewer 2 Report
The reviewed article deals with important problems in the field of sustainable development, i.e. the influence of climatic and anthropogenic factors on the hydrological regime of a tropical river with free flow. The extensive statistical analysis as well as the amount of data used and bibliographic items are impressive. This proves that researchers are specialized in their area of ​​interest, as well as scientific reliability. The set goals and research questions are clear and consistently implemented throughout the article. The research is of a high standard, however, I would ask you to respond to the comments below.
1) What height is the research area? What is the percentage share of individual types of land use (this question is important due to the conclusions presented later)?
2) On the map (Figure 1), please use arrows to mark the direction of the rivers.
3) Do the data in Table 1 match all the data obtained? The text shows that they were available for each year, while in the table these are selected years. If these were annual data, why were these years included in the table? However, if these are all data, they only show a certain trend, and additionally they do not overlap, so the obtained results and conclusions could be incorrect.
4) I wonder if accepting only three ranges of correlation can be considered correct. I would suggest the following ranges for the strength of correlation relationships: <0.2 - weak correlation, 0.2-0.4 - low correlation, 0.4-0.6 - moderate correlation, 0.6-0.8 - high correlation, 0 , 8-0.9 - very high correlation, 0.9-1.0 - almost full correlation. In this case, the analysis can be carried out in more detail and conclusions can be drawn from it. I am aware that in this type of analysis, there may be many factors that will affect the final result, and there will also be random errors beyond your control.
5) Table 10 - please correct its appearance so that the data fits on one line (e.g. 100%, 11.598 etc.)
6) The conclusions could be more elaborate by citing specific values ​​of anthropogenic and climatic parameters on the hydrological conditions of the river (flow regime). Please sign the X axes where possible in charts.
7) Please verify some entries in the bibliography, e.g. are DOI numbers added everywhere, are the surnames written consistently (e.g. in item 6 they are written in capital letters), are all abbreviations expanded (e.g. item 28, 54), have they been translated into English names from another language (e.g. item 31). Please add the language used in publications in the items, if it is different than English.
The article will be eligible for publication in the Sustainability journal after considering the above observations.
Reviewer 3 Report
The manuscript proposed by Leo et al. “Anthropogenic and Climate Effects on a Free Dam Tropical River: Measuring the Contributions on Flow Regime” aims to quantify the effects of climate variation and human disturbance upon river dynamic investigating the Dam upstream in Petro river. The work is interesting, well written with a good English, grammar, and phrasing structure.
Here some general comment:
Please specify in the introduction other available methodologies suitable for the same purpose highlighting the main advantages of the chosen methods.
Please specify the novelty of the applied methodology because it seems the only motivation to apply this study was highlighted in line 77 and I think is not a sufficient motivation to justify this publication. I think is so crucial my previous recommendation to specify the main methodology’s advantages.
Study area need to be reworked. I suggest enlarging the land use description as the map is not enough to describe the cover near the considered dam. Moreover, there is any groundwater body? If yes, is it recharged by the river or vice-versa? Furthermore give an insight on the geological setting of the area.
Please explain better the component of equation 1.
Also, the explanation of equation 2 and 3 need to be improved. You wrote that Pi is the annual average precipitation (mm) in the year i. Assuming that I’m considering the year 2000, what is the annual average? Do you mean total? And again, Pm express the mean annual…it refers to a time period? Please explain.
In Discussion can you imagine new elaboration and modification that has not been considered in this approach?
Organize the conclusion remarking the main funding maybe using a bullet point.
Round 2
Reviewer 3 Report
The authors reviewed the manuscript taking into account the reviewers' suggestions. The manuscript has now improved significantly